# Antibody Response to the SARS-CoV-2 Vaccine and COVID-19 Vulnerability during the Omicron Pandemic in Patients with CLL: Two-Year Follow-Up of a Multicenter Study

**DOI:** 10.3390/cancers15112993

**Published:** 2023-05-30

**Authors:** Francesca R. Mauro, Diana Giannarelli, Clementina M. Galluzzo, Andrea Visentin, Anna M. Frustaci, Paolo Sportoletti, Candida Vitale, Gianluigi Reda, Massimo Gentile, Luciano Levato, Roberta Murru, Daniele Armiento, Maria C. Molinari, Giulia Proietti, Sara Pepe, Filomena De Falco, Veronica Mattiello, Luca Barabino, Roberta Amici, Marta Coscia, Alessandra Tedeschi, Corrado Girmenia, Livio Trentin, Silvia Baroncelli

**Affiliations:** 1Hematology, Department of Translational and Precision Medicine, Sapienza University, 00161 Rome, Italy; mc.molinari91@gmail.com (M.C.M.); giulia.proietti@uniroma1.it (G.P.); pepe@bce.uniroma1.it (S.P.); 2Design and Analysis of Clinical Trials Unit, Scientific Directorate, IRCS Fondazione Policlinico Universitario A. Gemelli, 00168 Rome, Italy; diana.giannarelli@gmail.com; 3National Center for Global Health, Istituto Superiore di Sanità, 00161 Rome, Italy; clementina.galluzzo@iss.it (C.M.G.); roberta.amici@iss.it (R.A.); silvia.baroncelli@iss.it (S.B.); 4Hematology and Clinical Immunology Unit, Department of Medicine, University of Padua, 35122 Padova, Italy; andrea.visentin@unipd.it (A.V.); livio.trentin@unipd.it (L.T.); 5ASST Grande Ospedale Metropolitano Niguarda, 20162 Milan, Italy; annamaria.frustaci@ospedaleniguarda.it (A.M.F.); alessandra.tedeschi@ospedaleniguarda.it (A.T.); 6Institute of Hematology and Center for Hemato-Oncology Research, University of Perugia and Santa Maria della Misericordia Hospital, 06129 Perugia, Italy; paolo.sportoletti@unipg.it (P.S.); filomena.defalco@unipg.it (F.D.F.); 7Department of Molecular Biotechnology and Health Sciences, University of Torino and Division of Hematology, A.O.U. Città della Salute e della Scienza di Torino, 10125 Torino, Italy; candida.vitale@unito.it (C.V.); marta.coscia@unito.it (M.C.); 8Hematology Unit, Fondazione IRCCS Cà Granda Ospedale Maggiore Policlinico, 20122 Milan, Italy; gianluigi.reda@hotmail.it (G.R.); veronica.mattiello@policlinico.mi.it (V.M.); 9Hematology Unit AO of Cosenza, Cosenza and Department of Pharmacy, Health and Nutritional Sciences, University of Calabria, 87036 Rende, Italy; massim.gentile@tiscali.it; 10Department Hematology-Oncology, Azienda Ospedaliera Pugliese-Ciaccio, 88100 Catanzaro, Italy; leluc13@alice.it; 11Hematology and Stem Cell Transplantation Unit, Ospedale Oncologico A. Businco, ARNAS “G. Brotzu”, 09134 Cagliari, Italy; roberta.murru@tiscali.it; 12Unit of Hematology, Stem Cell Transplantation, University Campus Bio-Medico, 00128 Rome, Italy; d.armiento@unicampus.it; 13Department of Medical Sciences and Public Health, University of Cagliari, 09124 Cagliari, Italy; luca.barabino@hotmail.com; 14Azienda Policlinico Umberto I, 00161 Roma, Italy; girmenia@bce.uniroma1.it

**Keywords:** chronic lymphocytic leukemia, COVID-19, Omicron

## Abstract

**Simple Summary:**

We prospectively analyzed COVID-19 morbidity and severity in 200 consecutive patients with CLL. Increased COVID-19 morbidity was observed in vaccinated patients with CLL. With a median follow-up of 23.4 months, 41% of patients experienced COVID-19. Most patients, 36.5%, experienced COVID-19 during the Omicron pandemic, 26% of patients were hospitalized, and 4% died. Moreover, 10% had re-infections. In multivariate analysis of elderly patients, *TP*53 disrupted, heavily pre-treated, and those in early treatment with targeted agents showed increased vulnerability to COVID-19. Given the persistent risk of infection due to the continuous emergence of SARS-CoV-2 variants, our results support the importance of new vaccines and protective measures to prevent and mitigate COVID-19 in patients with CLL.

**Abstract:**

High morbidity and mortality due to COVID-19 were described in the pre-vaccination era in patients with chronic lymphocytic leukemia (CLL). To evaluate COVID-19 morbidity after the SARS-CoV-2 vaccine, we carried out a prospective study in 200 CLL patients. The median age of patients was 70 years; 35% showed IgG levels ≤ 550 mg/dL, 61% unmutated IGHV, and 34% showed *TP*53 disruption. Most patients, 83.5%, were previously treated, including 36% with ibrutinib and 37.5% with venetoclax. The serologic response rates to the second and third dose of the vaccine were 39% and 53%, respectively. With a median follow-up of 23.4 months, 41% of patients experienced COVID-19, 36.5% during the Omicron pandemic, and 10% had subsequent COVID-19 events. Severe COVID-19 requiring hospitalization was recorded in 26% of patients, and 4% died. Significant and independent factors associated with the response to the vaccine and vulnerability to COVID-19 were age (OR: 0.93; HR: 0.97) and less than 18 months between the start of targeted agents and vaccine (OR: 0.17; HR: 0.31). *TP*53 mutation and ≥two prior treatments also emerged as significant and independent factors associated with an increased risk of developing COVID-19 (HR: 1.85; HR: 2.08). No statistical difference in COVID-19 morbidity was found in patients with or without antibody response to the vaccine (47.5% vs. 52.5%; *p* = 0.21). Given the persistent risk of infection due to the continuous emergence of SARS-CoV-2 variants, our results support the importance of new vaccines and protective measures to prevent and mitigate COVID-19 in CLL patients.

## 1. Introduction

During the early stages of the severe acute respiratory syndrome coronavirus 2 (SARS-CoV-2) pandemic in 2020, before the vaccination strategy, patients with hematologic malignancies paid a high toll in terms of rates of patients with severe and fatal cases of coronavirus disease 2019 (COVID-19) [1,2,3]. Given the impaired humoral and cellular immunity due to the disease and the immunosuppressive effects of treatments, significant morbidity and mortality rates were recorded in patients with chronic lymphocytic leukemia (CLL) in 2020 in the early phases of the pandemic [4,5,6,7]. Impaired immunity was also responsible for suboptimal responses to the SARS-CoV-2 vaccine in CLL patients [8,9,10,11,12,13,14]. The global use of the SARS-CoV-2 vaccines and the emergence of mutated variants have modified the current pandemic scenario. Little is known about the long-term benefit of the vaccination strategy in patients with CLL and how the spread of new, highly transmissible variants impacted this patient population. We carried out a multicenter study in vaccinated patients with CLL to prospectively evaluate the morbidity and severity of symptomatic COVID-19 during the pandemic’s different phases, including the Omicron pandemic.

## 2. Patients and Methods

### 2.1. Patients and Methods

During the SARS-CoV-2 pandemic, the main hematology centers of our country were invited to participate in this study during an online meeting and ten agreed to participate. This study included 200 consecutive CLL patients who were vaccinated against SARS-CoV-2 between 1 February 2021, and 16 July 2021.

Eligibility criteria included CLL diagnosis [15], age ≥18 years, no clinical nor serologic signs of prior SARS-CoV-2 infection, and at least two doses of the BNT162b2 mRNA SARS-CoV-2 vaccine, the first vaccine authorized in Italy and widely available at vaccination centers.

The primary endpoint of this study was to evaluate the incidence, severity, and mortality of COVID-19 in vaccinated patients with CLL. The secondary endpoints were factors predicting the serologic response to the vaccine, the impact of the humoral response, and baseline clinical factors on the occurrence of COVID-19.

Clinical observation started from the first dose of the vaccine. The database was locked on 15 March 2023.Clinical characteristics of patients were extracted from medical records. They included demographic data, baseline characteristics, and treatment of CLL (lymphocyte count, Rai stage, immunoglobulin levels, beta-2microglobulin, disease status, prior treatment, number and type of previous treatments, IGHV and *TP*53 mutational status, *TP*53 deletion). Data about COVID-19 severity, management (hospitalization, home care), treatment, and outcome were also recorded.

Patients received at least two BNT162b2 mRNA SARS-CoV-2 vaccine doses three weeks apart. A third dose of the same vaccine was offered after at least three months from the second dose. Patients were informed about the importance of preventive measures, early symptom of COVID-19, and the need for immediate diagnostic screening. COVID-19 diagnosis was based on the positivity of respiratory specimens for SARS-CoV-2 by PCR testing in patients with symptomatic infection.

A centralized assessment of the antibody response was made at the Istituto Superiore di Sanità (I.S.S.) of Rome. Blood samples were taken before the first dose of the vaccine and three weeks after the second and third doses. Additional samples were taken six months from the second dose of the vaccine to evaluate the persistence of IgG antibodies to the SARS-CoV-2 virus. Details about methods to assess COVID-19 IgG levels are reported in the Appendix A.

### 2.2. Statistical Analysis

We analyzed the impact of the following variables on the serologic response to the second dose of the SARS-CoV-22 vaccine: CLL duration, gender, age (<70 vs. ≥70 years), CIRS (<6 vs. ≥6), IgG levels (<550 vs. ≤550 mg/dL), lymphocytes count (<5 vs. ≥5 × 10^9^/L), beta-2microglobulin (<3.5 vs. ≥3.5mg/dL), Rai stage (0–II vs. III–IV), progressive disease at the time of vaccination (present vs. absent), prior treatment (present vs. absent), number of previous treatments (0 + 1 vs. ≥2), IGHV (mutated vs. unmutated), deletion and, or mutation of *TP*53 (present vs. absent), the interval between last rituximab administration and the SARS-CoV-2 vaccine (≤12 months vs. >12 months or never given). In the cohort of patients who received targeted agents, in addition to these variables, we also tested the type of inhibitor administered and the interval (<18 vs. ≥18 months) between the start of treatment and the SARS-CoV-2 vaccine. The clinical and serologic characteristics of patients who developed COVID-19 were also analyzed. Moreover, we evaluated the impact of the same variables tested to identify risk factors associated with the development of COVID-19. Survival curves were calculated according to the Kaplan and Meier method. The Cox regression model was implemented to estimate the Hazard Ratio (H.R.) for each patient’s characteristics. In multivariable analysis, we tested only items with a *p* < 0.10 at the univariate evaluation. The same approach was used for assessing the impact of tested variables on the serological response (positive vs. negative); in this case, the logistic regression model was implemented to evaluate Odds Ratios (OR). Confidence intervals (C.I.s) were calculated at the 95% level. Unless otherwise specified, a *p*-value of less than 0.05 was considered significant. All analyses were performed using the IBM SPSS v.21.0 statistical software (IBM Corp., Armonk, NY, USA).

This study has been carried out according to ethical and scientific quality standards of Good clinical practice (GCP) and approved by the Ethics Committees of the participating centers. Patients were asked to sign an informed consent.

## 3. Results

### 3.1. Baseline Characteristics of Patients

Patient disposition is summarized in Appendix A, and baseline characteristics of patients are described in Table 1.

The median follow-up of patients from the first dose of the SARS-CoV-2 vaccine was 23.4 months (range, 4.5–25.7). The median age of patients was 70 years (range, 38–90), 34.5% had a CIRS ≥6, and 35% had IgG levels ≤550 mg/dL. Rai stage III–IV was present in 8% of patients, and clinical signs of active disease in 16%. Unmutated *IGHV* was recorded in 61% of the cases, and *TP*53 deletion and, or mutation in 34%. Details about prior or ongoing treatment are reported in Table 2.

Most patients, 167 (83.5%), were previously treated. Twenty patients (10%) were in remission after front-line chemoimmunotherapy, and 147 (73.5%) were previously treated with targeted agents. Among them, 72 (36%) patients were on an ibrutinib-based treatment (front-line therapy, 21%; advanced-line, 15%), and 75 (37.5%) had received a venetoclax-based therapy (front-line, 13.5%, advanced-line, 24%) including 10 in remission and off treatment after front-line venetoclax + rituximab. The median time between vaccination and the start of chemoimmunotherapy was 54.5 months (range, 9–210 months), and 18 months for targeted therapy (ibrutinib-based treatment, 36 months; venetoclax-based therapy, 13 months). A total of 135 (77.5%) patients had been previously exposed to rituximab, including 33 (16.5%) within one year before the first dose of the vaccine.

### 3.2. Serological Response and Factors Predicting the Serologic Response to the Vaccine

The serologic response was analyzed in 195 patients while five were excluded (high levels of anti-SARS-CoV-2 antibodies at baseline, 3; one dose only of the vaccine, 2). A total of 76 of 195 (39%) patients developed an adequate antibody response after the second dose of the vaccine with a median titer of 90 Au/mL (range, 5.87–160 Au/mL) anti-SARS-CoV-2 IgG. The rates of serologic response according to treatment were: treatment-naive patients, 69%; remission after chemoimmunotherapy, 65%; ibrutinib, 26%; venetoclax, 32% (Figure 1). A serologic response was recorded in 3/33 (9%) patients who had the last administration of anti-CD20 monoclonal antibodies within 12 months before vaccination.

Age (OR: 0.93 [95% CI: 0.90–0.96] *p* < 0.0001), IgG levels (OR: 0.28 [95% CI: 0.13–0.58], *p* = 0.001), and the time between the previous administration of CD20 monoclonal antibodies and vaccination (OR: 0.10 [95% CI: 0.03–0.37], *p* = 0.001), had a significant and independent impact on the serologic response in multivariable analysis, (Appendix A). The analysis was then restricted to the 143 patients who received targeted therapy. No serologic responses were recorded in patients with progressive disease at the time of the vaccine. The type of targeted agent, ibrutinib or venetoclax, did not impact response, while the interval (<18 vs. ≥18 months) between the start of treatment and the vaccine revealed an independent impact on the serologic response (OR: 0.17 [95% CI: 0.06–0.44], *p* < 0.0001) together with age (OR: 0.96 [95% CI: 0.92–0.99], *p* = 0.038) and IgG levels (OR: 0.31 [95% CI: 0.12–0.79], *p* = 0.014) (Appendix A). A total of 182 patients received the third dose of the same vaccine after a median time of six months from the second dose (range, 3–9 months), and 165 were tested for the serologic response. A response was detected in 52% of patients tested after the third dose of the vaccine (Figure 1). A total of 27 of the 103 (26%) patients who had failed to respond to the second dose responded to the booster dose, while 76 (74%*)* maintained seronegativity. Even with the limit of patients not tested after the third dose, we could count at least 103 (76 + 27; 53%) patients who developed an antibody response to two or three doses of the vaccine, while 92 (119–27; 47%) maintained seronegativity. The response to the third dose of the vaccine according to previous treatment is described in Figure 1.

### 3.3. Clinical Characteristics and Outcomes of Patients with COVID-19

As of 1 March 2023, 80 (41%) patients included in this study have developed COVID-19. For patients who reported typical signs of COVID-19 and required home care, the diagnosis was validated by RT-PCR positivity for SARS-CoV-2 certified by a microbiology laboratory. For hospitalized patients, the diagnosis was based on discharge reports. The 24-month COVID-19-free survival rate was 56.3%, (Figure 2). The clinical characteristics of patients who experienced COVID-19 are summarized in Table 3 and Figure 3.

COVID-19 was diagnosed in one patient (1/80, 1%; 1/195, 0.5%) before July 2021, when the alpha variant was dominant in our country, in 8 (8/80, 10%; 8/195, 4%) between July and December 2021, during the delta pandemic, and in 71 (71/80, 89%; 71/195, 36.5%) and after January 2022 when Omicron variants became dominant. Eight (10%) patients experienced subsequent COVID-19 events during the Omicron pandemic (two COVID-19 events, seven patients; three COVID-19 events, 1).

The proportion of infected patients previously treated with ibrutinib or venetoclax was similar, 39% and 36%. All patients but four had received at least three doses of the vaccine (range 2–5), including 12 patients who had four and three who received five doses. Most patients (76/80; 95%) interrupted targeted agents at the time of COVID-19. Mild COVID-19-related symptoms were reported by 59 (74%) patients who received home care only. Twenty-one (26%) of patients with COVID-19 showed a severe infection requiring supplemental oxygen and hospitalization, including nine who needed care in intensive care units.

COVID-19 treatment varied widely according to the severity of the infection, local treatment protocol, and time of diagnosis. Treatment consisted of anti-inflammatory agents with or without antibiotics in 16 (20%) patients. Anti-viral drugs or anti-SARS-CoV-2 monoclonal antibodies were given to 35 (44%) and 14 (17.5%) patients. In addition, hospitalized patients received oxygen, antibiotics, dexamethasone, and nine mechanical ventilation. Home care could not be defined in 15 (19%) cases.

A total of 555 of the 60 (92%) patients on targeted therapy at the time of COVID-19 resumed targeted therapy. Three (3/200; 1.5%; 3/80, 4%) patients died due to COVID-19 during the Omicron pandemic. The three dead patients were previously treated with chemoimmunotherapy and were on active treatment with venetoclax at the time of COVID-19. Moreover, they had received three vaccine doses without developing an antibody response and showed an additional factor of increased frailty for severe COVID-19 (age > 70 years, hypogammaglobulinemia).

### 3.4. Risk Factors of COVID-19

The same baseline variables tested for their impact on the serologic response were also analyzed for their effect on the risk of developing COVID-19. In multivariate analysis, three baseline factors emerged as significant and independent risk factors for developing COVID-19, age (HR: 0.97 [95% CI: 0.95–0.9995], *p* = 0.046), the number of prior treatments, ≥2 vs. 1 (HR: 2.08 [95% CI: 1.27–3.40], *p* = 0.004), the presence of *TP*53 deletion and, or mutation (HR: 1.85 [95% CI: 1.06–3.25], *p* = 0.032), (Appendix A). When the analysis was restricted to patients treated with targeted agents, the same factors maintained a significant independent impact on the risk of developing COVID-19 together with the interval, ≥ 18 vs. <18 months, between vaccination and the start of prior targeted therapy (HR:0.31 [95% CI: 0.15–0.63], *p* = 0.001) (Appendix A).

COVID-19 was diagnosed in 42/92 (46%) patients who failed to respond to the last dose of the vaccine, and in 38/103 (37%) patients who developed an antibody response. The difference in the incidence of COVID-19 between seronegative and seropositive patients did not reach significance (*p* = 0.21).

In February 2022, the Italian Medicines Agency (AIFA) authorized pre-exposure prophylaxis of the SARS-CoV-2 infection with a 150/150 mg dose of two long-acting anti-Spike monoclonal antibodies, tixagevimab/cilgavimab, in immunocompromised patients regardless the response to the prior anti-SARS-CoV-2 vaccine. Fifty (62.5%) patients included in this study received pre-exposure prophylaxis with tixagevimab/cilgavimab, 34 patients without a prior COVID-19 event (COVID19-free patients) and 16 with a previous event (COVID-19 experienced patients). COVID-19 was diagnosed in 19 (56%) COVID-19-free patients (including 13 seropositive patients) after a median of 2.75 months (range, 0.25–11) from the tixagevimab/cilgavimab administration. No COVID-19-related deaths were recorded in these patients, but 8 (8/19, 42%) required hospitalization. Patients who had received tixagevimab/cilgavimab after a prior COVID-19 event did not experience subsequent COVID-19 events.

## 4. Discussion

This multicenter study provides a prospective long-term scenario about the morbidity, severity, and mortality of COVID-19 during the different phases of the pandemic in patients with CLL who received the SARS-CoV-2 vaccine.

Data from the first analysis of this study showed a serological response to two doses of the anti-SARS-CoV-2 vaccine in only 39% of patients. Other studies have described suboptimal responses in patients with similar characteristics [8,9,10,11,12,13,14]. Moreover, we also observed lower rates of responses in patients treated with ibrutinib or venetoclax, 26%, and 31.5%, respectively [8,9,10,11,12,13,14,16,17,18]. This finding was not surprising as low humoral responses to other vaccines were also described in ibrutinib-treated patients [19,20,21]. Interestingly, a longer time, ≥18 months, between the start of ibrutinib or venetoclax and vaccination was associated with a better chance of developing an antibody response. This finding suggests that more prolonged treatment with targeted agents could produce better clinical and humoral responses. As described by Herishanu et al. [22], the third dose of the vaccine promoted an antibody response in 26% of patients with stable disease who had failed to respond to the second dose.

We evaluate with a two-year follow-up the protective impact of the vaccine and the humoral response to the vaccine in terms of COVID-19-related morbidity and mortality.

During the first pandemic, two doses of the BNT162b2 vaccine conferred protection against COVID-19 in 94–95% of healthy subjects [23,24]. Despite the low rate of serologic responses, 99% of vaccinated patients included in this study remained COVID-19-free during the first pandemic dominated by the alpha variant. This observation further indicates that antibody titers should not be considered an absolute benchmark for the adequate immune protection of vaccines [25]. Interestingly, Shen et al. detected a T-cell response to a SARS-CoV-2 peptide pool in approximately 80% of vaccinated patients, regardless of their antibody response, suggesting a role of cellular immunity promoted by the vaccine [16].

In line with the Italian epidemiological data [26], we recorded a rapid increase in COVID-19 cases during the Omicron pandemic, with 36.5% of vaccinated patients experiencing COVID-19. This study’s prospective design led us to capture all symptomatic COVID-19 events, reducing the bias of selecting severe cases only. However, the proportion of asymptomatic infections could not be estimated as serology was not assessed in all patients beyond the second dose of the vaccine. The increased rate of COVID-19 cases during the Omicron pandemic is a well-known epidemiological data representing the result of the antibody escape by variants carrying mutated spike proteins. The immune escape developed by mutant variants raises concern about the efficacy of antibodies induced by prior vaccination or infection [27,28,29]. In this study, the proportion of patients who developed COVID-19 with or without a serologic response to the vaccine was not statistically different. About half of the patients who developed COVID-19 had an antibody response to the prior vaccine. In addition, 9% of patients with COVID-19 experienced a subsequent COVID-19 event.

A detrimental effect of mutated variants is the decreased efficacy of monoclonal antibodies designed on previous variants and developed for therapeutic or prophylactic use.

During the Omicron pandemic, 19/34 (56%) COVID-free patients who received pre-exposure prophylaxis tixagevimab/cilgavimab developed a not-fatal COVID-19 but eight required hospitalization.

The continuous spreading of new mutant variants will likely expose, despite vaccination, immunocompromised subjects to recurrent and potentially severe COVID-19.

We found that older age and two or more prior treatments emerged as significant and independent factors associated with an increased risk of developing COVID-19. In addition to being associated with reduced antibody responsiveness, less than 18 months of treatment with venetoclax or ibrutinib appeared to be associated with a higher vulnerability to COVID-19. This finding again suggests the impact on immunological reconstitution produced by more prolonged treatment with targeted agents. Surprisingly, a *TP*53 disruption also showed a significant and independent effect on an increased risk of COVID-19. The increased vulnerability of patients with *TP*53 disruption to symptomatic SARS-CoV-2 infection could be explained by recent data suggesting the role of p53 in anti-viral immunity [30]. Mice lacking the *TP*53 gene die due to malignancies and approximately one-quarter due to infection. It has been argued that p53 could exert anti-viral immunity in response to the viral DNA or RNA by triggering apoptosis of infected cells [30,31]. In this way, the apoptosis of the infected cells may limit the infection of neighboring cells. In addition, the increased risk of symptomatic infections could also be due to the role played by p53 in pathways, including factors involved in the anti-viral response, such as IFN-1 [32].

During the early pandemic, over 80% of non-vaccinated patients with CLL who developed COVID-19 were hospitalized, and a third died [4,5].

Despite the lack of effective antibodies against spike-mutant variants, we recorded a relatively low rate of patients who required hospitalization and fatal COVID-19, 26% and 4%, respectively. This finding is in line with the consensus that Omicron variants are highly transmissible, but COVID-19 is less severe among vaccinated people [33,34,35]. The lower intrinsic virulence of mutated variants, the availability of better therapeutic approaches, and the more limited use of steroids have played a relevant role in improving the outcome of COVID-19also in vaccinated patients with hematologic diseases, including those with CLL [36,37,38,39,40,41,42]. Interestingly, Parry et al. [17] showed that cellular immunity induced by prior vaccines produced responses against Omicron variants equivalent to those seen against the ancestral strain.

In our study, 26% of vaccinated developed severe COVID-19 requiring hospitalization. Although during the Omicron pandemic, fatal COVID-19 are rare, the potential severity of COVID-19 in immunocompromised subjects should not be underestimated, particularly in elderly and actively treated patients with hematologic disorders, including CLL [36,37,38,39]. Compared to other observational studies, including those that enrolled patients from our country, our prospective study had a long follow-up extended to the Omicron pandemic. The results of our study provided new insights into the impact of clinical, biologic and treatment characteristics on the long-term outcomes of vaccinated patients with CLL.

Our data show an increased vulnerability to COVID-19 in CLL patients during the Omicron pandemic, associated with age, *TP*53 disruption, prior treatment, and earlier phase of treatment with targeted agents. Moreover, COVID-19 was clinically severe in a not negligible proportion of patients. Given the persistent risk of infection due to the continuous emergence of SARS-CoV-2 variants, our results support the importance of new vaccines and protective measures to prevent and mitigate COVID-19 in patients with CLL.

## Figures and Tables

**Figure 1 cancers-15-02993-f001:**
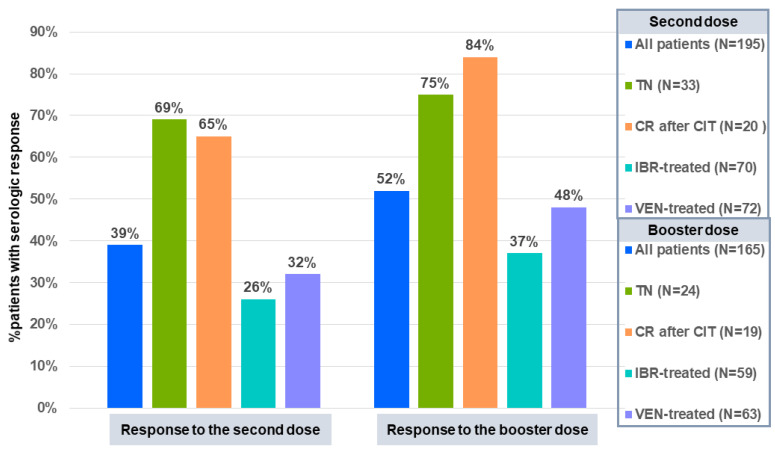
Serologic response of patients with CLL to the SARS-CoV-2 vaccine by treatment. Abbreviations: CLL, chronic lymphocytic leukemia, SARS-CoV-2, severe acute respiratory syndrome coronavirus 2; TN, treatment naïve; CR, complete response; CIT, chemoimmunotherapy; IBR, ibrutinib; VEN, venetoclax.

**Figure 2 cancers-15-02993-f002:**
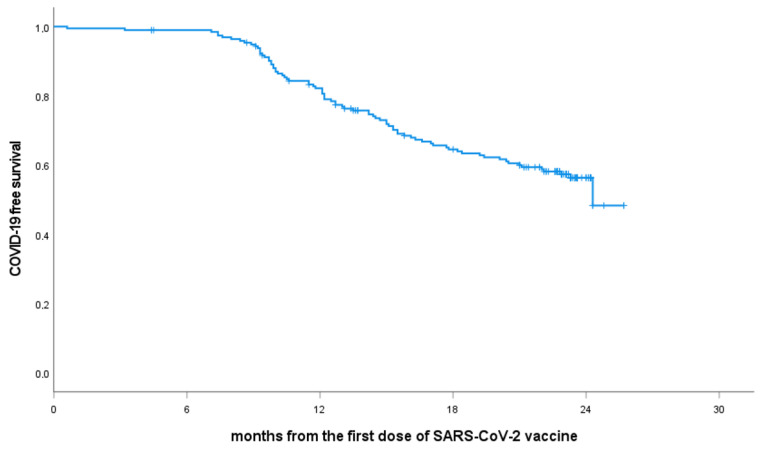
COVID-19-free survival from the SARS-CoV-2 vaccine. Abbreviations: COVID-19, Coronavirus disease 2019; SARS-CoV-2, severe acute respiratory syndrome coronavirus 2.

**Figure 3 cancers-15-02993-f003:**
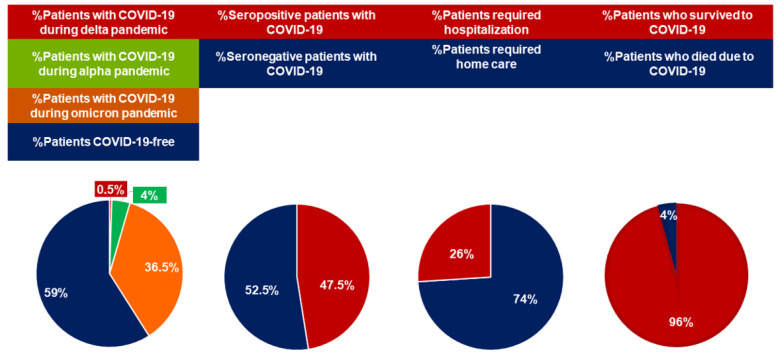
Epidemiologic and clinical characteristics of COVID-19 in vaccinated patients with CLL. Abbreviations: COVID-19, Coronavirus disease 2019; CLL, chronic lymphocytic leukemia.

**Table 1 cancers-15-02993-t001:** Baseline characteristics of patients at the time of the first dose of the SARS-CoV-2 vaccine.

	N = 200 (%)
Median age at the time of the anti-SARS-CoV-2 vaccine, years (range, IQR)	70 (38–90) (61–76)
Sex M/F	113/87
Median time from CLL to the anti-SARS-CoV-2 vaccine, months, (range)	92 (1–387)
Median follow-up from the first dose of anti-SARS-CoV-2 vaccine, months, (range)	23.4 (4.5–25.7)
CIRS ≥ 6	69 (34.5)
IgG levels ≤ 550 mg/dL	70 (35)
Median lymphocyte count ×10^9^/L, (range)	3.12 (0.4–218.0)
Beta2-microglobulin ≥ 3.5 mg/dL	34/176 (19.3)
Rai stage III–IV	16 (8.0)
Clinical signs of progressive disease	32 (16.0)
Del13q	31/175 (17.7)
Tris 12	29/175 (16.6)
Del 11q	30/175 (17.1)
Del17p	34/175 (19.4)
*TP*53 mutation	50/179 (27.9)
Del17p and/or *TP*53 mutation	58/172 (34)
IGHV unmutated	111/183 (61)
IGHV mutated	72/183 (39)

Abbreviations: Ig, immunoglobulins; CIRS, Cumulative Illness Rating Scale; Del, deletion; Tris, trisomy; *TP*53, tumor protein p53; IGHV, immunoglobulin heavy chain variable region mutations.

**Table 2 cancers-15-02993-t002:** Prior and ongoing treatments at the time of the first dose of the SARS-CoV-2 vaccine.

	N = 200 (%)
The median number of prior treatments (range)	1 (0–8)
Treatment naïve patients	33 (16.5)
Previously treated patients	167 (83.5)
Front-line chemoimmunotherapy only	20 (10.3)
Targeted agents	147 (73.5)
Ibrutinib-based treatment ^(1)^	72 (36)
Front-line	42 (21.0)
Ibrutinib single agent	28 (14)
ibrutinib + rituximab	14 (7)
Advanced-line Ibrutinib ^(1)^	30 (15.0)
Venetoclax-based treatment	75 (37.5)
Front-line venetoclax + rituximab ^(2)^	27 (13.5)
Advanced-line venetoclax ± rituximab	48 (24.0)
Venetoclax + rituximab ^(3)^	21 (10.5)
Venetoclax single agent	27 (13.5)
Median number of months (range) between vaccine and start of:	
Chemoimmunotherapy	54.5 (9–210)
Ibrutinib or venetoclax–based treatment	18 (3–90.5)
Ibrutinib-based treatment	36 (2–90.3)
Venetoclax-based treatment	13 (0.5–44)
Patients previously treated with rituximab	135 (77.5)
Last rituximab within 12 months before vaccination	33 (16.5)
Last rituximab more than 12 months before vaccination	102 (51–0)

^(1)^ Ongoing ibrutinib-based treatment: 72/72 patients. ^(2)^ Ongoing treatment with front-line venetoclax + rituximab: 17/27 patients, median time from venetoclax + rituximab discontinuation: 12.5 months (range, 5–17 months). ^(3)^ Ongoing treatment with advanced-line venetoclax+ rituximab, 21/21 patients.

**Table 3 cancers-15-02993-t003:** Clinical characteristics and outcomes of patients who developed COVID-19.

	**N = 80 (%)**
Median age, years (range)	69.1 (39–89)
CIRS ≥ 6	20 (25)
IgG levels < 550 mg/dL	27 (34)
Rai stage III–IV	5 (6)
Unmutated IGHV	41 (51)
*TP*53 mutation/deletion	15 (19)
More than one COVID-19 event ^(1)^	8 (10)
Treatment naive	14 (17.5)
Prior treatment	66 (82.5)
Chemoimmunotherapy only	6 (7.5)
Ibrutinib based	31 (39)
Venetoclax based ^(2)^	29 (36)
Last rituximab administration within six months	3 (4)
Number of doses of the vaccine before COVID-19	
Two	4 (5)
Three	61 (76)
Four	12 (15)
Five	3 (4)
Known serologic response to the last dose of the vaccine	
Absent	42 (52.5)
Present	38 (47.5)
Pre-exposure prophylaxis with tixagevimab/cilgavimab	50 (62.5)
COVID-19-free patients	19/34 (56)
COVID-19-experienced patients	0/16 (0)
Pandemic phase of COVID-19 diagnosis ^(3)^	
Before July 2021	1 (1)
Between July and December 2021	8 (10)
From January 2022	71 (89)
Discontinued ibrutinib or venetoclax-based treatment during COVID-19	76/80 (95)
Clinical management of COVID-19	
Home care	59 (74)
Hospitalized [No patients who required intensive care unit]	21 (26%) [9]
Treatment	
Anti-inflammatory agents ± antibiotics	16 (20)
Anti-viral agents	35 (44)
Monoclonal antibodies	14 (17.5)
Not specified	15 (19)
COVID-19-related deaths	3 (4)
Recovered from COVID-19 and resumed targeted therapy	74/80 (92.5)

Abbreviations: COVID-19, Coronavirus disease 2019; SARS-CoV-2, severe acute respiratory syndrome coronavirus 2; Ig, immunoglobulins; *TP*53 gene, tumor protein p53 gene; Del., deletion; IGHV, immunoglobulin heavy chain variable region gene. ^(1)^ Two COVID-19 events, 6 patients; three COVID-19 events, 1. ^(2)^ Eight patients with CR and treatment-free from a median number of 8.5 months from venetoclax + rituximab discontinuation. ^(3)^ SARS-CoV-2 variants dominant in our country: alpha variant before July 2021, delta variant between July and December 2021, and omicron variants from January 2022.

## Data Availability

De-identified individual data supporting the presented findings will be available upon reasonable request. Proposals for access should be sent to mauro@.bce.uniroma1.it.

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
