# Peer review of "Antibody Response to the SARS-CoV-2 Vaccine and COVID-19 Vulnerability during the Omicron Pandemic in Patients with CLL: Two-Year Follow-Up of a Multicenter Study"

_cancers, 2023, doi:10.3390/cancers15112993_

Round 1

Reviewer 1 Report

Thank you for this intersting research article

I do have some concerns you might solve before publication. 

The title does not  fit to the main focus of the paper, serological response after first or second vaccine dose. 

As you mention it is a prospective study: nowadays they are usually registered in a clinical study registry

It remains unclear when the data presented in table 1 were collected: first dose of vaccine? second dose of vaccine?

In table 1, it remains unclear why median time from first vaccine is 16.5 months, although all the other data seem to be from timepoint at first or second dose?

It remains unclear why median follow-up time after first vaccination was 23 months (fulltext), but 16.5 months in table 1 

It remains unclear when data collection for final analyis took place

It remains unclear, how SARS-CoV2 infection was assessed: by physicians (RT-PCR), patient records, or self-.reported (especially for patients not hospitalised)?  

It remains unclear, how many participants from the whole population received fourth or fifth dose of vaccine (only reported for participants that developed COVID-19)

chapter 3.3: please be consistent whether you evaluated SARS-CoV2 infection or COVID-19

results from table 4: when have clinical data been assessed? At first dose of vaccine?

Reviewer 2 Report

In this manuscript authors describe COVID-19 in CLL patients in the middle of the COVID-19 pandemic. The manuscript is informative and well written. However, it could benefit if certain aspects are considered.          

-          Please, include references to previous works from big consortia (in Europe, for example EPICOVIDEHA) that support your results, not only single centre experiences

-          Please, describe how and where were the centres included into analysis

-          Since the data are almost 2 years old, it might be beneficial to express this directly in the title, with something like “xxxxx, February-July 2021”

-          “Age 18 years” means that all patients were that age. If “age >= 18 years”, please correct

-          Could authors provide a reason why only patients receiving BNT162b2 and not any of the other treatment options were included to analysis?

-          The manuscript would benefit, and be complete, after the inclusion of a paragraph regarding which statistical methods were used for result obtention

-          Line 109: median follow up until when? (check throughout the whole manuscript, please)

-          Gender (sociology) and sex (biology) are not equivalent. If the authors are describing sex (biology), please check it

-          Figures: Please, include the meaning of the abbreviations used. Figures should be self-explanatory without the help of the main text

-          Why would authors repeat the same content in table 3 and suppl. table 1? Authors may prefer to include suppl. table 1 to table 3, for the sake of clarity and to avoid data duplication

-          Figure 3 can be a bit misleading. All pies have the same size, but they represent a different n

-          Why are supplementary file and non-published file the same?

-          Suppl. fig. 1: What denominator is “COVID19-NON RELATED DEATHS: 6 (6/195,3%)”. Please, check this figure, making it coherent (regarding capital letter use [no need for that] or bold letters)

-          Suppl. table 1: “Months from the start of Ibrutinib or venetoclax-based therapy ≥ 18 vs. <18” is NOT significant: the OR contains the 1. Please, check

-          In Suppl. table 2: How can age OR 95% CI include the 1, and still be “sufficiently” significant?
